# Environment-sensitive turn-on fluorescent probe enables live cell imaging of myeloperoxidase activity during NETosis

Enebie Ramos Cáceres ⓘ , Lotte Kemperman & Kimberly M. Bonger ⓘ ✉

Myeloperoxidase (MPO) plays an important role in the immune response of human neutrophils and has been implicated in autoimmune conditions, cardiovascular disorders, and neurodegeneration. Current methods to detect MPO activity rely on the detection of HOCl using activatable probes or require challenging experimental procedures. Therefore, these tools provide limited information about the dynamics and localization of MPO in complex molecular processes such as NETosis in real time. In this study, we report a ''turn-on'' activity-based probe that fluoresces exclusively upon binding to MPO, exhibits minimal background fluorescence in buffered aqueous media, and is blocked by MPO inhibitors. Our probe facilitates real-time imaging of direct MPO activity in human neutrophils and HL-60-derived granulocytes during NETosis under wash-free conditions. Furthermore, it allows for the discrimination between different triggers of NETosis in human neutrophils. These findings hold promise for advancing our understanding of the role of MPO in immune responses and inflammatory conditions.

Myeloperoxidase (MPO) is a heme peroxidase with strong antimicrobial properties prominently found within the primary granules of human neutrophils[1–3]. Upon neutrophil activation, NAPDH oxidase produces superoxide radicals that quickly dismutate to $H_2O_2$[2–4]. Active MPO utilizes $H_2O_2$ in its halogenation and peroxidation cycles to generate hypochlorous acid (HOCl) and other reactive oxygen species (ROS), which aid in pathogen neutralization during phagocytosis and degranulation[5,6]. As powerful oxidants, HOCl and other ROS can oxidize most biomolecules, which could lead to cell death and tissue damage when they are produced in large quantities[7–9]. Consequently, MPO is considered as pathogenic in contexts outside of the innate immune response, including autoimmune diseases[10], cardiovascular diseases[11], and neurodegeneration[12–14]. Therefore, it is critical to be able to detect MPO activity both in vitro and in vivo to understand its role in pathological processes.

Besides its role in phagocytosis, neutrophil-derived MPO participates in neutrophil extracellular trap (NET) formation, a process better known as NETosis. NETs are comprised of DNA in the form of decondensed chromatin decorated with histones and various proteins, including MPO[15]. NETosis can be induced by a variety of stimuli. For example, interleukin 8 (IL-8), bacterial lipopolysaccharides (LPS), and small molecules such as ionomycin, a calcium ionophore, or phorbol 12-myristate 13-acetate (PMA), a well-known protein kinase C (PKC) activator[16]. Although the exact molecular mechanisms of NET formation remain unclear, the enzymes involved in the process depend highly on the stimulus[16–18].

Interestingly, the precise role of MPO in NET formation is a subject of debate. Metzler et al. suggested that MPO mediates the release of neutrophil elastase from primary granules upon PMA-induced neutrophil activation[19]. Then, neutrophil elastase translocates to the nucleus to cleave histones and assist in chromatin decondensation. These findings are consistent with the impaired ability of neutrophils from $MPO^{-/-}$ humans to generate NETs upon PMA stimulation[20]. However, while MPO activity is necessary for PMA-induced NETosis, inhibition of MPO had no effect on NET formation when neutrophils were triggered by bacteria[21]. Therefore, new ways to detect MPO activity could help to unravel the precise role of MPO and its contribution in the process of NET formation and overall immune response.

In recent years, fluorescence-based probes have become valuable tools to image a wide range of biological subjects due to their small size, tunable fluorescent properties, and high signal-to-background ratios. Current fluorescence-based probes to detect MPO activity are indirect and focus on its main enzymatic product, HOCl. For example, Shepherd et al. developed a

Department of Synthetic Organic Chemistry, Institute for Molecules and Materials, Radboud University, Heyendaalseweg 135, 6525AJ Nijmegen, The Netherlands. ✉e-mail: k.bonger@science.ru.nl

near-infrared fluorescent probe to detect HOCl in stimulated human neutrophils and MPO[+] macrophages[22]. Similarly, the groups of Liu et al. and Tian et al. developed other fluorescent probes to image HOCl in resting and $H_2O_2$-stimulated HL-60 cells, respectively[23,24]. Although HOCl production is exclusive to MPO, HOCl itself is highly reactive, very short-lived, and diffuses rapidly. Recently, Wang et al. developed a series of MPO-activatable hydroxy-indole sensors that were used for the in vitro and in vivo detection of MPO activity. In here, MPO-mediated oxidation of the hydroxyl function followed by reaction of the resulting radical with closeby aromatic residues resulted in covalent labeling of proximal proteins. An additional biotin motif in the probes allowed visualization of MPO activity in mice models through biotin-streptavidin interactions and MRI agents[25,26]. Although together these tools are very valuable, they provide limited information about the precise live-cell dynamics and localization of MPO in complex molecular processes such as NETosis.

In an effort to develop tools to study active MPO directly, Ward et al. synthesized the first activity-based probe against MPO containing a 2-thioxanthine inhibitor linked to an alkyne tag[27]. With this probe, the authors were able to label MPO directly on the catalytic heme modality in the active site and in complex lysates in a $H_2O_2$-dependent manner. Inspired by their work, we designed and evaluated a series of environment-sensitive fluorescent activity-based probes by equipping 2-thioxanthine with the solvatochromic fluorophores 4-N,N-dimethylamino-1,8-napthalimide (4-DMN), 4-(N,N-dimethylaminosulfonyl)-7-amino-2,1,3-benzoxadiazole (DBD), and 7-amino-4-nitro-2,1,3-benzoxadiazole (NBD) as depicted in Scheme 1. Environment-sensitive fluorophores have gained attention as useful reporters in strategies targeting hydrophobic structures and hydrophobic ligand-binding domains[28,29]. Importantly, these fluorophores display high fluorescence in hydrophobic environments and have low fluorescence intensity in aqueous media. These properties ensure a "turn-on" character that greatly reduces background signal allowing wash-free live-cell imaging (Scheme 1A). In recent years, such fluorophores have been used to study many proteins including peptoid helix structures[30], membrane-bound receptors[31], and ion channels[32]. The solvatochromic 'turn-on' probes that we present herein allow imaging of MPO activity in the process of NETosis in both fixed cells and live cells in real-time wash-free conditions.

## Results and Discussion

2-Thioxanthine derivatives have been reported as mechanism-based inhibitors of MPO that modify the heme group of the enzyme[27,33]. They undergo a one-electron oxidation in the presence of MPO to generate a free radical that could be released into the solution, or form a covalent bond between the methyl group of the heme pyrrole ring and their sulfur atom[27,33].

To obtain probes 1-6 (Scheme 1B), we synthesized two 2-thioxanthine scaffolds containing an ethyl and diethyl ether spacer (E1 and E2) to which we could attach different environment-sensitive fluorophores (Scheme 1B, C). We hypothesized that spacer length could have a significant impact on the properties of the probes where long spacers could make the fluorophore more solvent-exposed leading to lower fluorescence intensities while shorter spacers could restrict the binding of the inhibitor to the enzyme. Subsequently, amine-containing scaffolds E1 and E2 were coupled to 4-DMN anhydride, DBD-F, or NBD-Cl to produce probes 1-6 (Scheme 1C).

To validate the solvatochromic properties of the probes, we recorded the fluorescence excitation and emission spectra of probes 1-6 in a 10 µM solution in $CH_2Cl_2$, $CH_3CN$, $CH_3OH$, and PBS buffer (pH 7.4, Table 1, Supplementary Table 1). These solvents have different polarities thereby emulating the hydrophobic enzyme-probe binding site environment (Supplementary Figs. 1-6). Probes 1 and 2 displayed ~300-fold fluorescence increase in $CH_2Cl_2$ compared to PBS buffer (pH 7.4). This effect was less prominent for 3 and 4 with a ~110-fold fluorescence increase, and for 5 and 6 with a ~30-fold fluorescence increase. In addition, all probes displayed a spectral red-shift in both fluorescence excitation and emission maxima that increased with solvent polarity, a characteristic typical of solvatochromic dyes.

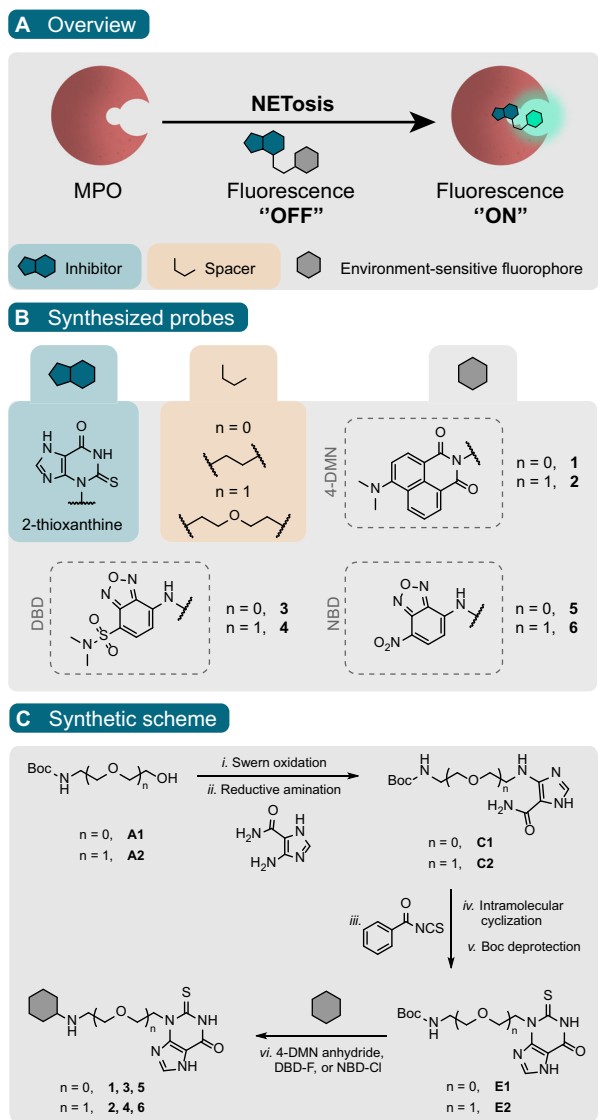

Scheme 1 | A Schematic illustration of the environment-sensitive fluorescent probes for selective detection of MPO activity. The mechanism is based on the binding of the covalent inhibitor to the active site pocket of the enzyme, whereby the adjacent hydrophobic environment can cause the environment-sensitive fluorophore to exhibit stronger fluorescence. In the absence of MPO, the fluorescent probe has low fluorescence. B Chemical design and formulas of fluorescent probes 1-6 for MPO. 2-Thioxanthine serves as a covalent inhibitor scaffold to which different environment-sensitive fluorophores are conjugated through an ethyl (n = 0) or diethyl ether spacer (n = 1). Probes 1, 2 are conjugated to the 4-DMN fluorophore. Probes 3, 4 and 5, 6 are conjugated to the structurally similar DBD and NBD fluorophores, respectively. C Synthetic scheme of fluorescent probes 1-6.

Having obtained the fluorescent properties of probes 1-6, we determined their functional properties by the Amplex Red assay as previously reported (Table 1, Supplementary Figs. 1-6)[27]. In this assay, 10-acetyl-3,7-dihydroxyphenoxazine is oxidized by MPO in the presence of $H_2O_2$ to resorufin, a highly fluorescent compound with an fluorescence excitation and emission maxima of 571 and 585 nm, respectively. Fortunately, probes 1, 2, 4, 5, and 6 retained similar $IC_{50}$ values as those previously reported for 2-thioxanthine-based inhibitors despite the presence of the environment-sensitive fluorophores[27,33]. However, we noticed a slight negative trend in the $IC_{50}$ of probes containing the shorter ethyl spacer. This effect was more evident in probe 3, which experienced a ~4-fold reduction in $IC_{50}$ values compared to its counterpart probe 4.

**Table 1 | Spectroscopic and functional properties of probes 1-6**

| Probe | $\lambda_{ex}$ (nm) | | $\lambda_{em}$ (nm) | | $IC_{50}$ (µM)[a] |
|---|---|---|---|---|---|
| | PBS (pH 7.4) | CH$_2$Cl$_2$ | PBS (pH 7.4) | CH$_2$Cl$_2$ | |
| 1 | 457 | 415 | 560 | 510 | 0.854 ± 0.044 |
| 2 | 446 | 415 | 560 | 512 | 0.479 ± 0.017 |
| 3 | 435 | 417 | 600 | 532 | 4.573 ± 1.835 |
| 4 | 435 | 420 | 600 | 535 | 1.088 ± 0.151 |
| 5 | 473 | 452 | 558 | 522 | 0.497 ± 0.086 |
| 6 | 488 | 452 | 558 | 517 | 0.388 ± 0.035 |

[a]$IC_{50}$ was obtained from Amplex Red assay (**1-5**, $n = 5$; **6**, $n = 4$, $t = 30$ min).

With probes **1-6** in hand, we opted for a cell-based screening strategy to find suitable probes to image MPO activity during NETosis (Fig. 1A). In this approach, we stimulated freshly isolated human polymorphonuclear neutrophils (hPMN) with PMA and incubated with probes **1-6** at concentrations near their $IC_{50}$ for 3 h. We hypothesized that at these concentrations MPO would not be completely inhibited and the process of NETosis should be minimally affected. Subsequently, probes that show a significant ''turn-on'' fluorescence compared to unstimulated cells are selected and tested for colocalization with an anti-MPO antibody in fixed cells. With this cell-based strategy, potential suitable probes to image MPO activity during NETosis can be immediately identified. Therefore, live hPMN were incubated with PMA and probes **1-6** and imaged directly (Fig. 1B). The results show that of these probes, **1** and **2** displayed the highest fluorescence intensity 3 h after induction of NETosis. Moreover, probe **1** displayed higher signal-to-background ratio (S/B) and more than double the mean fluorescent intensity (MFI) of probe **2** under identical imaging conditions. These findings suggest that probe **1** could serve as a promising ''turn-on'' probe for the detection of MPO activity during NETosis.

To verify the ''turn-on'' and binding character of probe **1** as well as its use for the detection of MPO activity, we incubated probe **1** with commercially available purified MPO using 4-aminobenzoic hydrazide (ABAH), a well-known myeloperoxidase inhibitor[34], and BSA as controls (Fig. 1C). Under identical reaction conditions, probe **1** incubated with MPO displayed a ~ 13-fold fluorescence increase compared to probe **1** alone, ABAH pre-treatment of MPO, or an identical protein concentration of BSA. In addition, to confirm irreversible inhibition and functional properties similar to parent compounds[27], we calculated the $k_{inact}$, $K_I$, and $k_{inact}/K_I$ values for probe **1** to be 0.0046 ± 0.0001 s$^{-1}$, 1.079 ± 0.073 µM, and 4263 M$^{-1}$s$^{-1}$, respectively (Supplementary Fig. 1). Furthermore, we confirmed the chemical stability of probe **1** in cell culture media with or without serum for 4 and 24 h at 37 °C (Supplementary Fig. 7). The results suggest that probe **1** is chemically stable under these conditions.

Next, we determined the specificity of probe **1** for MPO using ABAH and colocalization analysis with an anti-MPO antibody in both HL-60-derived granulocytes (dHL-60) and hPMNs. dHL-60 cells naturally express MPO and have emerged as a potential ''cultured neutrophil'' cell line[35]. Their ability to undergo respiratory burst upon stimulation with PMA makes them an attractive alternative to hPMN, as these cells should be isolated from fresh blood and challenging to work with in non-clinical settings. To obtain dHL-60 cells, we differentiated HL-60 cells in cell culture medium containing 1.25% DMSO or 1 µM ATRA for 5–6 days[35]. After validating cell viability, the differentiation process was confirmed by nuclear morphology assessment (Supplementary Fig. 8). To validate dHL-60 cells as potential mimics for neutrophils, we measured their NET release and respiratory burst capabilities upon stimulation with PMA and ionomycin (Supplementary Fig. 8). We found that, similar to hPMN, both DMSO- and ATRA-dHL-60 cells respond to PMA. However, PMA-stimulated ATRA-dHL-60 cells release more NETs and produce more ROS than DMSO-dHL-60 cells (Supplementary Fig. 8). Thus, we show herein that ATRA-differentiated dHL-60 cells qualify as suitable hPMN mimics to study MPO in PMA-induced NETosis.

In fixed cells, both hPMN and dHL-60 cells incubated with PMA and probe **1** at 1 µM displayed a significant signal decrease when cells were pre-treated with ABAH. In addition, in both cell types probe **1** and anti-MPO signal generally showed similar staining and colocalization (Fig. 1D, E, Supplementary Fig. 9). Spatial signal correlation was highlighted by the significant overlap of normalized signal intensity profiles recorded across the chosen representative vector (Fig. 1F, Supplementary Fig. 9). Colocalization was assessed and quantified over a 2.6 µm Z-stack using Pearson's correlation coefficient (PCC), Mander's split coefficients (M1 and M2) and Van Steensel's cross-correlation function (CCF) (Supplementary Fig. 10)[36]. For hPMN and dHL-60 cells, Van Steensel's CCF peaked at dx 1 and 0 with a CCF of 0.74 and 0.77, respectively. These values were consistent with both PCC (0.74, 0.77) and Mander's split coefficients (hPMN, M1 = 0.66, M2 = 0.74; dHL-60, M1 = 0.79, M2 = 0.75) and are together representative of strong colocalization[36]. Strikingly, probe **1** and the anti-MPO antibody colocalized predominantly in the periphery of NETotic cells, not on NETs themselves (Fig. 1E, Supplementary Fig. 11). It is worth noting that MPO binds NETs released during NETosis and for this reason it is a commonly used NETosis marker in immunofluorescence studies. This is consistent with our immunofluorescence observations of MPO being present in NETs across multiple experiments. However, MPO might have other functions other than its canonical role in the innate immune response, varying from its involvement in several signal transduction pathways to non-enzymatic functions, many of which have been compiled in a recent review by Van-hamme et al. [37]. There is also evidence suggesting that MPO undergoes extensive post-translational processing during neutrophil maturation[28]. Since probe **1** is present for the whole duration of NETosis, we believe that the MPO present in these NETs could either be inactive, denatured, or some form of immature pre-processed enzyme. Investigating the nature of this form of MPO further could help to better understand the molecular mechanisms of NETosis and other endogenous functions of MPO in neutrophil biology.

To highlight the use of probe **1** for live-cell imaging, we incubated live hPMN and dHL-60 with probe **1** and PMA for 3 h and imaged in wash-free conditions (Fig. 2A). By design, environment-sensitive fluorophores could fluoresce in other hydrophobic environments such as the phospholipid bilayer of the plasma membrane or in lipid droplets (e.g. Nile Red). Fortunately, probe **1** did not display fluorescence in these structures (Fig. 2A, B). Importantly, ABAH pre-treatment significantly reduced probe **1** signal in both hPMN and dHL-60 cells (Fig. 2A, B). Similar to fixed cells, probe **1** signal concentrated in the cell periphery of NETotic cells and did not colocalize with DNA.

Finally, probe **1** was also used to differentiate between NETosis induced by PMA or ionomycin. As mentioned previously, the molecular mechanisms leading to NETosis are highly stimulus-dependent[16–18]. PKC activation through PMA leads to the overproduction of $H_2O_2$ by NAPDH oxidase needed for MPO activity. Ionomycin and other calcium ionophores, on the other hand, cause a rapid increase of intracellular $Ca^{2+}$ concentration, which produces mitochondrial ROS and activates peptidyl arginine deiminase 4 (PAD4), a key enzyme driving chromatin decondensation. While MPO activity is important for PMA-induced NETosis, its relevance has been debated in other forms of NETosis[21]. We hypothesized that probe **1** could be used to differentiate between PMA and ionomycin, as PMA has been shown to induce a quicker and stronger oxidative burst than calcium ionophores[18,38], thereby leading to stronger MPO activation. To prove our hypothesis, live hPMNs and dHL-60 cells were incubated with probe **1** and PMA or ionomycin for 3 h (Fig. 2C). The results show that of these, only PMA-treated hPMNs displayed a high fluorescence intensity of probe **1**, despite both presenting a similar amount of NETotic cells. Probe **1** signal in ionomycin-stimulated cells was not significantly different from unstimulated background levels (Fig. 2C, D). Together, these results show that probe **1** could be used as an imaging tool to rapidly monitor MPO activity in live cells undergoing NETosis in a stimulus-specific manner. In the future, it would be interesting to use probe **1** to investigate MPO activity in hPMN with other stimuli such as cytokines,

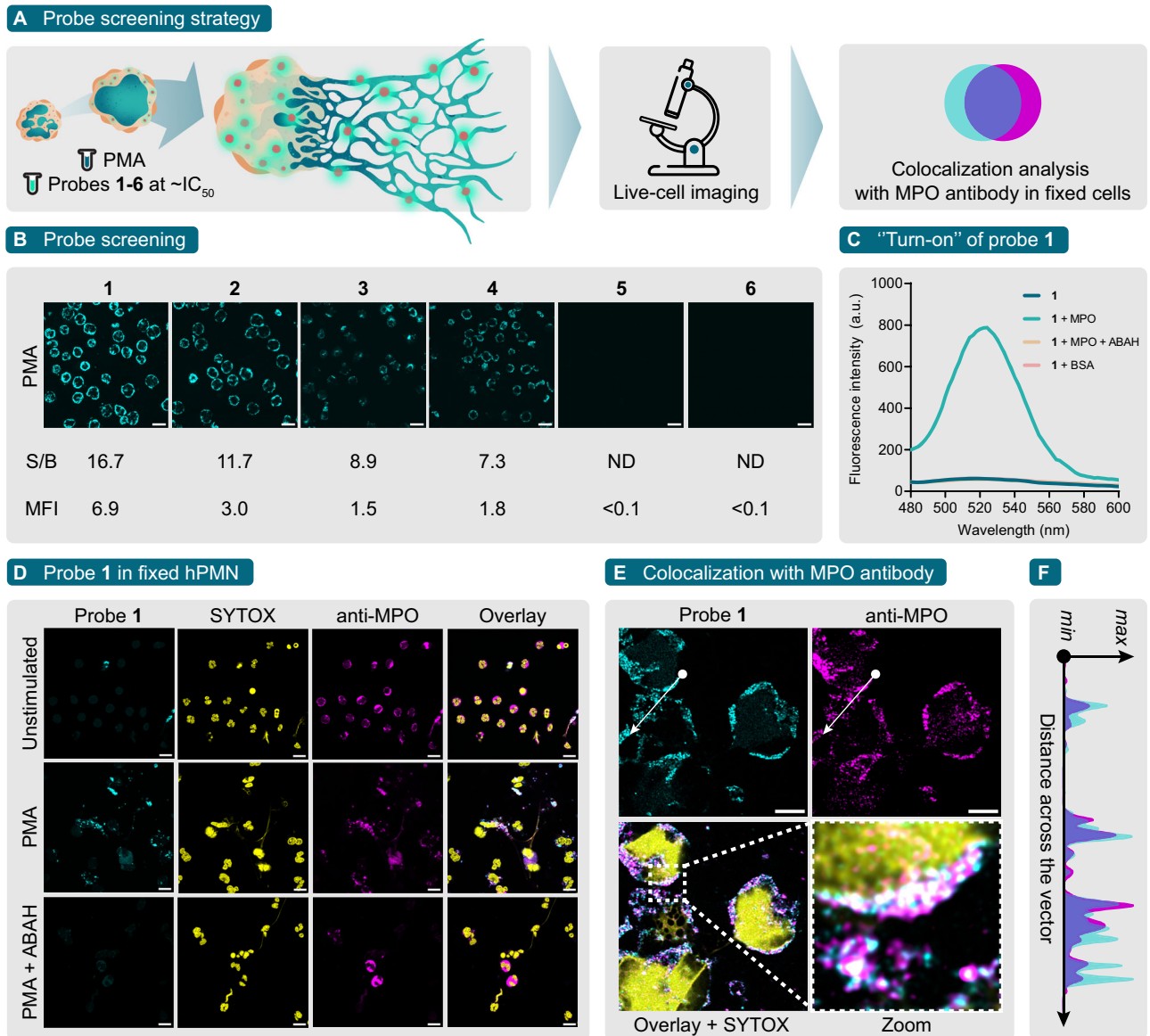

**Fig. 1 | Probe screening, testing, and verification with MPO and hPMN. A** Our probe screening strategy design to find suitable "turn-on" probes for MPO activity during NETosis. Live hPMN are induced into NETosis by PMA and imaged in the presence of probes **1-6** at concentrations near their $IC_{50}$ value without washing. Suitable probes are then validated in live cells and in fixed cells through colocalization analysis with MPO antibody. **B** CLSM panel of live hPMN stimulated with 100 nM PMA in the presence of probes **1-6** (cyan) at concentrations near their $IC_{50}$ for 3 h at 37 °C, 5% $CO_2$. S/B is the signal-to-background ratio for each probe and MFI is the mean fluorescence intensity in gray values obtained from whole images. **C** Fluorescence "turn-on" emission spectra of probe **1**. Probe **1** fluorescence emission spectrum was measured alone (blue), with 15 μM MPO (teal), with 15 μM MPO

pre-treated with 500 μM ABAH (yellow), or with 15 μM BSA (pink) in PBS (pH 7.4) in the presence of 150 μM $H_2O_2$. **D** CLSM panel of fixed hPMN stimulated with PMA in the presence of 1 μM probe **1** (cyan) for 3 h at 37 °C, 5% $CO_2$. Where appropriate, cells were pre-treated with 100 μM ABAH for 30 min before the addition of probe **1**. NETs were detected with SYTOX Orange (yellow). MPO was detected by immunostaining with a primary anti-human MPO antibody and a secondary goat-anti human antibody conjugated to AF647 (magenta). **E** Colocalization of probe **1** and anti-human MPO antibody in hPMN stimulated with PMA. **F** Histogram overlap (purple) of probe **1** (cyan) and anti-MPO antibody (magenta) normalized intensity as a function of the distance across the vector in (**E**). Scale bar = 10 μm. ND: not determined.

bacteria, or yeast. Finally, the spectral properties of probe **1** make it a suitable target for tissue microscopy techniques such as two-photon microscopy[39]. Therefore, in addition to the in vitro detection of MPO, probe **1** could potentially be applied to image MPO-dependent inflammatory conditions in vivo and ex vivo.

In summary, we have developed an environment-sensitive fluorescent activity-based probe to visualize MPO activity in real time. We have demonstrated its applicability in both fixed and live hPMN and dHL-60 cells undergoing NETosis. Finally, the probe serves as a powerful tool for distinguishing between PMA- or ionomycin-induced NETosis pathways in living cells, which could be broadened to other kinds of stimuli.

## Methods

### Synthesis of fluorescent probes 1-6

Detailed information about the synthesis and characterization of the synthetic intermediates and probes can be found in the Supplementary Information and Supplementary Data 1 and 2.

### Measurement of spectroscopic properties of fluorescent probes 1-6

Probes **1-6** were prepared into 10 mM DMSO stock solutions and diluted to 10 μM with $CH_2Cl_2$, $CH_3CN$, $CH_3OH$, or PBS (pH 7.4). The fluorescence excitation spectra of each probe was measured between 300 and 480 nm for

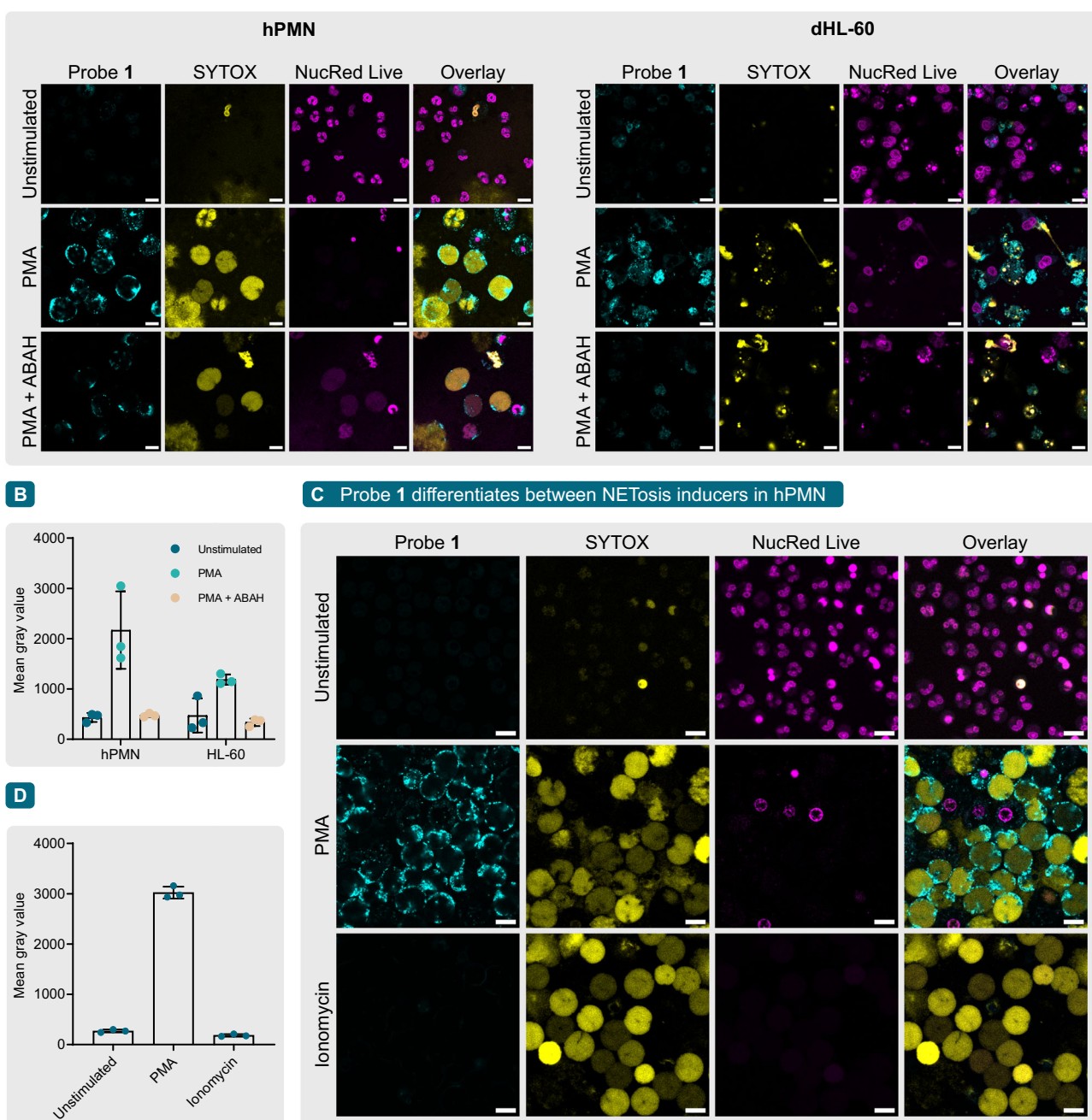

**Fig. 2 | Live-cell imaging of MPO activity in hPMN and dHL-60 cells in wash-free conditions. A** CLSM panel of live hPMN and dHL-60 cells stimulated with 100 nM PMA in the presence of 1 μM probe **1** (cyan) for 3 h at 37 °C, 5% $CO_2$. Where appropriate, cells were pre-treated with 100 μM ABAH for 30 min before the addition of probe **1**. **B** Mean gray value ($n$ = 3 independent experiments) of probe **1** signal in (**A**). **C** CLSM panel of live hPMN stimulated with PMA or 4 μM ionomycin in the presence of 1 μM probe **1** for 3 h at 37 °C, 5% $CO_2$. **D** Mean gray value ($n$ = 3 independent experiments) of probe **1** signal in (**C**). In all cases, NETs were detected with SYTOX Orange (yellow) and the DNA of live cells was stained with NucRed Live (magenta). All images are representative of three independent experiments. Scale bar = 10 μm. Data is presented as mean ± SD.

probes **1-4** and 300 and 500 nm for probes **5** and **6**. The fluorescence emission spectra of each probe was measured between 480 and 700 nm for probes **1-4** and 500 and 700 nm for probes **5** and **6**.

### Determination of functional properties of fluorescent probes 1-6 through Amplex Red assay

MPO peroxidase activity and the determination of inhibitor potencies were performed as described before with minimal changes[25]. In brief, to 78 uL of

assay mixture (PBS (pH 7.4), 1 μM $H_2O_2$, and 20 μM Amplex Red) were added 2 μL of probe **1-6** in concentrations from 0 to 40 μM in DMSO (final DMSO concentration of 2%) and mixed for 1 min. The reactions were initiated by the addition of 20 μL 500 pM MPO (final MPO concentration of 100 pM) followed by mixing for 5 s. Fluorescent changes (RFU/s) were monitored at room temperature every 20 s for 30 min with excitation and emission filters set at 530 and 580 nm, respectively. Initial rates ($V_0$) were determined using linear regression analysis of the first 600 s corresponding

to the linear range of the DMSO control. To calculate $IC_{50}$ values, $V_0$ and probe concentration were fit to a dose-response curve with a standard Hill slope of -1 using nonlinear regression analysis. The $k_{obs}$, $k_{inact}$, and $K_I$ values for probe **1** were calculated exactly as described before[27].

### Fresh human polymorphonuclear leukocytes isolation

Human polymorphonuclear leukocytes were isolated from fresh blood of healthy donors exactly as described before[38], in accordance with the Declaration of Helsinki, and under approval of the MREC Oost-Nederland (local registration number 2023-16405). Donor recruitment was performed according to all relevant guidelines and fresh blood was obtained with informed consent. In brief, 10 mL of fresh peripheral blood was carefully layered over 10 mL of Lymphoprep followed by centrifugation for 20 min, 400 *x g* at room temperature without acceleration or break. The neutrophil/erythrocyte layer was harvested, diluted with 5% FCS/HBSS (without $Ca^{2+}$/$Mg^{2+}$) and the erythrocytes sedimented with 1.5% dextran/0.45% NaCl for 15 min at 37 °C. Residual erythrocytes were lysed with 20 mL Milli-Q for 28 s before restoring isotonic conditions with 20 mL 1.8% NaCl. The cells were washed once with 5% FCS/HBSS (without $Ca^{2+}$/$Mg^{2+}$) and resuspended in RPMI 1640 without phenol red supplemented with 2% FCS and L-glutamine. Cell viability was determined using a trypan blue dye exclusion assay and the cell density adjusted to $2 \times 10^6$ cells/mL. Cell viability was routinely > 95%.

### Live-cell imaging

Cells were allowed to settle for 30 min on poly-L-lysine-coated glass-bottom 8-well Ibidi slides ($3 \times 10^5$ cells per well) in medium with or without 100 μM ABAH. To stimulate NETosis and visualize MPO activity, 100 nM PMA or 4 μM ionomycin was added along with probes **1-6** at concentrations ~$IC_{50}$, or alternatively at 1 μM for probe **1**, from a 10 mM DMSO stock and the cells were incubated for 3 h at 37 °C, 5% $CO_2$. After this time, one drop of NucRed Live 647 (ReadyProbes) and 1 μM SYTOX Orange were added to each well to visualize the DNA of live cells and NETs or the DNA of cells with a compromised plasma membrane, respectively.

### Immunofluorescence

Cells were allowed to settle for 30 min on poly-L-lysine-coated 12 mm coverslips ($3 \times 10^5$ cells per well) in medium with or without 100 μM ABAH. To stimulate NETosis and visualize MPO activity, 100 nM PMA was added along with 1 μM probe **1** from a 10 mM DMSO stock and cells were incubated for 3−4 h at 37 °C, 5% $CO_2$. Samples were fixed in 2% PFA in PBS (pH 7.4) for 20 min and the fixative quenched with 100 mM glycine in PBS (pH 7.4). Cells were permeabilized with 0.1% Triton X-100 in PBS (pH 7.4) and blocked with 5% normal goat serum in PBS (pH 7.4) for 30 min at room temperature. Samples were incubated with a primary rabbit anti-human MPO (EPR20257, Abcam, 1:400, 2 h to overnight, RT to 4 °C) antibody and a secondary goat anti-human antibody conjugated to AF647 (Abcam, 1:400, 1 h, RT). DNA was counterstained with SYTOX Orange. Colocalization analysis was performed over a 2.6 μm Z-stack with 0.06 μm steps.

### Image analysis

CLSM images were smoothed using the Smooth command in ImageJ and their brightness/contrast adjusted linearly (identically across directly comparable images) where appropriate for illustration purposes only. Colocalization analysis was performed with JaCoP[34]. An example of the threshold used for each slice of the Z-stack can be found in Supplementary Fig. 10. To quantify probe S/B, raw images were segmented using the Threshold command with Otsu's method. Then, the mean gray value of the signal within the threshold was divided by the mean gray value of the signal outside of the threshold. To quantify probe mean gray value (mean fluorescence intensity) in different conditions, the background of the raw images was subtracted using the Subtract Background (50 px) command and the mean gray value was determined without any segmentation using the Measure command.

## Data availability

The data that support the findings of this study are available from the corresponding author upon reasonable request.

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

## Acknowledgements
We would like to thank all the donors that participated in this study. We thank Dr. Daphne Dorst and Dr. Marije Koenders from the Department of Experimental Rheumatology at RadboudUMC for their contributions to isolating primary human neutrophils. This work is part of a project that has received funding from the European Research Council (ERC) under the European Union's Horizon 2020 research and innovation program (grant agreement No. 802940) and the NWO gravitation program 'Institute for Chemical Immunology' (NWO-024.002.009).

## Author contributions
E.R.C. and K.B. designed the project. E.R.C. and L.K. synthesized the molecular probes. E.R.C. performed all biological experiments. E.R.C. designed graphical contents and prepared the figures. E.R.C. and K.B. wrote and edited the manuscript. All authors have given permission for publishing of the manuscript.

## Competing interests
K.B. is a Guest Editor for *Communications Chemistry's* Covalent chemical probes Collection, but was not involved in the editorial review of, or the decision to publish this article. All other authors declare no competing interests.
