## [Transparent Peer Review file · Communications Chemistry]

Environment-Sensitive “Turn-On” Fluorescent Probe Enables Live Cell Imaging of Myeloperoxidase Activity during NETosis

Corresponding Author: Dr Kimberly Bonger

Version 0:

Reviewer comments:

Reviewer #1

(Remarks to the Author)

The paper reports a series of probes designed to bind Myeloperoxidase, via with a small molecule targeting ligand (which is a suicide inhibitor) – attached to a fluorophore. This is different to many fluorescent probes currently used for MPO analysis which use MPO activity to “decage” a fluorophore.

However, there is strong similarity to the known 5-hydroxyindole probes that have also been used as MPO labelling probes via suicide inhibition – e.g. via L-5-hydroxy-tryptophan-biotin/streptavidin AlexaFluor-647 e.g. the paper A versatile imaging platform with fluorescence and CT imaging capabilities that detects myeloperoxidase activity and inflammation at different scales.

This same entity (L-5-hydroxy-tryptophan) has also been used to irreversibly bind MPO for MRI studies (e.g. the paper Highly Efficient Activatable MRI Probe to Sense Myeloperoxidase Activity) and also has been used for PET imaging in vivo. As such it is important to compare the probe used here with this standard (5-hydroxy-tryptophan) and determine K_i and k_{inact} for both.

The probes are interesting in that they are analysed in CH_2Cl_2 , MeOH, DMSO and PBS. A question therefore relates to their solubility (in such a broad range of solvents) and how they were used/applied in cellular assays. Their solubilities should be determined/analysed. Do they aggregate at all? There is very little detail of how compounds are added to cells and the cellular assays themselves and how soluble they are. Table 1 has the ϵ ($M^{-1}cm^{-1}$) in water – but this is the solvent with the lowest fluorescence. This needs to be repeated with DMSO.

Cellular assays start with primary hPMN, but then seem to diverge and focus on the cell line HL-60 (differentiated). The initial studies on hPMN's should be repeated to keep it consistent. The data/images of probe 1 in hPMN and dHL-60 cells look quite different. There needs to be some co-staining with endosomal markers as it does seem to be very localised with the membrane and often quite punctate in nature. Do these cells undergo NETosis?

There needs to be some work on other protein binding partners for the probes. Albumin springs to mind as it has a number of binding pockets and it very non-selective in terms of its binding partners and is also highly abundant. The authors should run an ITC or fluorescence polarisation assay to measure its affinity and also fluorescence turn-on levels with some common proteins. I think this is also important as the plots show quite some co-staining.

Minor points:

The authors should perhaps think about the schematic they use for DNM, DBD and NBD. It matches DNM, but seems a little odd for the other two which are bicyclic, not tricyclic.

I would prefer the term spacer to linker. To me linker is usually a group that is cleavable, e.g. a linker used in solid-phase chemistry., whereas a a spacer is just that.

Reviewer #2

(Remarks to the Author)

Review manuscript Chemscomm.

Title: "Environment-Sensitive "Turn-On" Fluorescent Probe Enables Live Cell Imaging of Myeloperoxidase Activity during NETosis".

Authors: Enebie Ramos Cáceres, et al.

In this study, the authors designed and synthesized novel fluorescent probes for the detection of MPO in NETosis. This "turn-on" activity-based probe exhibited minimal background fluorescence in aqueous media, and was blocked by MPO inhibitors. The probe facilitated real-time imaging of direct intracellular MPO activity in human neutrophils and HL-60-derived granulocytes during NETosis under wash-free conditions.

The design of experiments is (very) good and the results support the conclusions.

Minor comments:

In the Introduction: Give both abbreviations for interleukin 8 (IL8, CXCL8), between brackets after its first use.

Major comments:

The authors state: "Remarkably, probes 1 and 2 displayed ~300-fold fluorescence increase in dichloromethane compared to PBS buffer". It may be that counter ions play a role, which are buffered in PBS and not in DCM. May be the authors can comment on this.

The affinity of the probes for MPO is not very high (low nanomolar range). How do the authors make sure that these probes/this probe will allow visualization of activated MPO in in vivo studies.

In the discussion, the authors mention that they are now testing the probe(s) in in vivo and ex vivo models. How stable are the compounds in PBS and against proteolytic degradation?

Version 1:

Reviewer comments:

Reviewer #1

(Remarks to the Author)

I am happy that the majority of there changes requested have been made.

However, although I am aware that Ki and kinact determinations are a lot of work, these are important for an irreversible/mechanism based inhibitor. The reason for this is that IC50 determinations are somewhat flawed when enzyme activity is being lost due to the inhibitor being activated and killing enzyme activity (this means here that IC50 values will vary depending on how long the enzyme and inhibitor have been mixed together - something that is irrelevant for other inhibitors) - and what is important is its binding affinity (Ki) and its rate of enzyme deactivation (kinact).

The authors also say "Having obtained the fluorescent properties of probes 1-6, we determined their functional properties by the Amplex Red assay as previously reported (Table 1, Supplementary Fig. 1-6). 27. Ref 27 also measures kinact and Ki values.

Reviewer #2

(Remarks to the Author)

According to this reviewer, the points of discussion have been addressed properly.

I support publication of the revised version.

Version 2:

Reviewer comments:

Reviewer #1

(Remarks to the Author)

what I asked for has now been done.

Reviewer #1

The paper reports a series of probes designed to bind Myeloperoxidase, via with a small molecule targeting ligand (which is a suicide inhibitor) – attached to a fluorophore. This is different to many fluorescent probes currently used for MPO analysis which use MPO activity to “degrade” a fluorophore.

However, there is strong similarity to the known 5-hydroxyindole probes that have also been used as MPO labelling probes via suicide inhibition – e.g. via L-5-hydroxy-tryptophan-biotin/streptavidin AlexaFluor-647 e.g. the paper A versatile imaging platform with fluorescence and CT imaging capabilities that detects myeloperoxidase activity and inflammation at different scales.

This same entity (L-5-hydroxy-tryptophan) has also been used to irreversibly bind MPO for MRI studies (e.g. the paper Highly Efficient Activatable MRI Probe to Sense Myeloperoxidase Activity) and also has been used for PET imaging in vivo. As such it is important to compare the probe used here with this standard (5-hydroxy-tryptophan) and determine K_i and k_{inact} for both.

>> We thank the reviewer for the comments and suggestion. We are aware of hydroxy-tryptophan molecules that covalently react with biomolecules upon oxidation by MPO. However, the mechanism of action for these molecules is different that with 2-thioxanthines. MPO oxidation of hydroxy-tryptophan results in phenoxy radical product that can diffuse from the MPO enzyme and react mainly with biomolecules in close proximity instead of with MPO itself. While these molecules are indeed very useful to image MPO activity with, for example, CT and MRI in vivo, they would not be useful for our studies to image MPO activity during NETosis in live cells in real time under wash-free conditions. Imaging in these conditions require molecules that bind to heme that resides in the active site of MPO and is well established for 2-thioxanthine inhibitors. As our approach involves the use of environment sensitive fluorophore that only fluoresces when the molecule is bound to the active site of MPO, we expect it unlikely that these ‘turn-on’ fluorophores work when the probes are bound to other distal enzymes. Therefore, and the fact that it would involve a large amount of synthesis, we do not see the additional value to determine K_i and K_{inact} for other targeting ligands of which mechanism is less clear. Nonetheless, we have discussed some work on these hydroxy-tryptophan molecules in the main text for completeness.

The probes are interesting in that they are analysed in CH₂Cl₂, MeOH, DMSO and PBS. A question therefore relates to their solubility (in such a broad range of solvents) and how they were used/applied in cellular assays. Their solubilities should be determined/analysed. Do they aggregate at all? There is very little detail of how compounds are added to cells and the cellular assays themselves and how soluble they are. Table 1 has the ϵ (M⁻¹cm⁻¹) in water – but this is the solvent with the lowest fluorescence. This needs to be repeated with DMSO.

>> We measured fluorescence of the probes in different solvents to validate and demonstrate the solvatochromic turn-on properties of the fluorophores. In addition, we prepared 10 mM stock solutions of all the probes in DMSO in which they are fully soluble as well as at micromolar concentrations in PBS (with 0.0001% DMSO) that we used for cell experiments.

Cellular assays start with primary hPMN, but then seem to diverge and focus on the cell line HL-60 (differentiated). The initial studies on hPMN's should be repeated to keep it consistent. The data/images of probe 1 in hPMN and dHL-60 cells look quite different. There needs to be some co-staining with endosomal markers as it does seem to be very localised with the membrane and often quite punctate in nature. Do these cells undergo NETosis?

>> The HL-60 cell line was included to further show that these cells, when differentiated, also work as a neutrophil model for PMA-induced NETosis in terms of MPO activity. With our probes, we demonstrate that hPMN and dHL-60 cells look different which is because dHL-60 cells respond less strongly than hPMNs to stimuli such as PMA and ionomycin. It is also more difficult to keep NETs on coverslips with these cells, as they are much more fragile and in lower abundance than those of hPMNs.

MPO is localized in primary granules; small, round structures of 100-200 nm that contain, in addition to MPO, proteases and other bactericidal molecules. We believe that the punctuate staining of probe 1 is due to the native MPO localization in hPMN.

Although we did not specifically look for it, we did not observe at any point the colocalization of probe 1 with the membrane in resting or stimulated hPMNs. Due to the nature of the environment-sensitivity of the fluorophores, any possible membrane association should be observable in resting hPMNs, but the probe exhibits low fluorescence under these conditions.

There needs to be some work on other protein binding partners for the probes. Albumin springs to mind as it has a number of binding pockets and it very non-selective in terms of its binding partners and is also highly abundant. The authors should run an ITC or fluorescence polarisation assay to measure its affinity and also fluorescence turn-on levels with some common proteins. I think this is also important as the plots show quite some co-staining.

>> We have addressed this point by measuring the fluorescence turn on of the probes with purified MPO and equal concentrations of BSA as control as described below.

The authors should perhaps think about the schematic they use for DNM, DBD and NBD. It matches DNM, but seems a little odd for the other two which are bicyclic, not tricyclic.

>> To avoid confusion, we have addressed this point and replaced the solvatochrome with a hexagon.

I would prefer the term spacer to linker. To me linker is usually a group that is cleavable, e.g. a linker used in solid-phase chemistry., whereas a a spacer is just that.

>> Spacer and Linker are both commonly used to connect two molecules. As we personally prefer the term 'linker' we kept it as is.

Reviewer #2

In the Introduction: Give both abbreviations for interleukin 8 (IL8, CXCL8), between brackets after its first use.

>> Addressed.

The authors state: "Remarkably, probes 1 and 2 displayed ~300-fold fluorescence increase in dichloromethane compared to PBS buffer". It may be that counter ions play a role, which are buffered in PBS and not in DCM. May be the authors can comment on this.

>> Although it is not shown, we have also measured the fluorescent properties in MilliQ, and it displays similar results to those with PBS. Further, the molecules are neutral and counter ions should not play a role.

The affinity of the probes for MPO is not very high (low nanomolar range). How do the authors make sure that these probes/this probe will allow visualization of activated MPO in in vivo studies.

>> Others have shown that 2-thioxanthine derivatives are capable of inhibiting MPO during inflammation in mice in the low micromolar range (<https://doi.org/10.1074/jbc.M111.266981>). Therefore, we expect similar or better results with our environment-sensitive activity-based probes, as they would be used in localized sites where neutrophils and MPO accumulate. Yet, the use of our MPO in vivo needs to be investigated and beyond the scope of this manuscript.

In the discussion, the authors mention that they are now testing the probe(s) in in vivo and ex vivo models. How stable are the compounds in PBS and against proteolytic degradation?

>> To address the stability of the probes in serum, we incubated the probes in cell culture medium supplemented with 2% FCS for 0, 4 and 24 h. The results show full chemical stability as measured by HPLC. We have included this data in the supplementary information. We expect the molecules to be stable to proteolytic degradation as there is no motif present in the molecules that can be cleaved by proteolytic enzymes.

Reviewer #1

However, although I am aware that K_i and k_{inact} determinations are a lot of work, these are important for an irreversible/mechanism based inhibitor. The reason for this is that IC_{50} determinations are somewhat flawed when enzyme activity is being lost due to the inhibitor being activated and killing enzyme activity (this means here that IC_{50} values will vary depending on how long the enzyme and inhibitor have been mixed together - something that is irrelevant for other inhibitors) - and what is important is its binding affinity (K_i) and its rate of enzyme deactivation (k_{inact}).

⇒ *We thank the reviewer for the comment and have now included the k_{inact}/K_i for the best performing compound 1. We calculated the k_{inact} , K_i , and k_{inact}/K_i values for probe 1 to be $0.0046 \pm 0.0001 \text{ s}^{-1}$, $1.079 \pm 0.073 \text{ }\mu\text{M}$, and $4263 \text{ M}^{-1}\text{s}^{-1}$ and addressed this in the manuscript and supplementary figure 1.*

Reviewer #2

According to this reviewer, the points of discussion have been addressed properly. I support publication of the revised version.

⇒ *We thank the reviewer for the comments and time for assessing this paper.*